# Effect of a single one-hour teaching session about environmental pollutants and climate change on the understanding and behavioral choices of adolescents: The BREATHE pilot randomized controlled trial

Yorusaliem Abrham[1,2☯], Siyang Zeng[1,2,3☯], Rachel Tenney[1,2,4], Caroline Davidson[1,5], Emily Yao[1,2], Chantal Kloth[1,2], Sarah Dalton[1,2], Mehrdad Arjomandi[1,2]*

1 Medical Service, San Francisco Veterans Affairs Health Care System, San Francisco, San Francisco, California, United States of America, 2 Department of Medicine, University of California, San Francisco, San Francisco, California, United States of America, 3 Department of Biomedical Informatics and Medical Education, University of Washington, Seattle, United States of America, 4 Department of Medicine, Weil Cornell University Medical Center, New York, New York, United States of America, 5 National Council for Mental Wellbeing, New York, New York, United States of America

☯ These authors contributed equally to this work.
* Mehrdad.arjomandi@ucsf.edu

**Data Availability Statement:** The datasets generated and/or analyzed during the current study

## Abstract

### Background

Despite the wealth of scientific information on the health effects of air pollution, the adult public's lifestyle continues to be largely detrimental towards the environment.

### Objective

The purpose of the study was to determine whether a short interactive teaching session on air pollution could shift reported behavioral choices of adolescents towards environmentally friendlier options.

### Methods

We performed a pilot randomized control trial in which eighth-grade students were randomized to receive a one-hour script-based teaching on either the effects of air pollution on lung health (intervention group) or the role of vaccination in public health (active control group). The enrolled students completed a survey (15 multiple-choice questions; five targeting understanding (score range 5 to 20); ten targeting behavioral choices (score range 10 to 38) newly designed for this study to evaluate their understanding and predict their future behavior towards air pollution immediately before, immediately after, and one month after the teaching sessions.

are available on Dryad at https://doi.org/10.7272/Q63T9FGB.

**Funding:** This work was supported by: 1. The California Tobacco-related Disease Research Program (T29IR0715 to MA). URL: https://www.trdrp.org/funded-research/ 2. National Library of Medicine Training Grant (NIH: T15LM007442 to SZ). URL: https://www.nlm.nih.gov The funders had no role in study design, data collection and analysis, decision to publish, or preparation of the manuscript. The statements and conclusions in this publication are those of the authors and not necessarily those of the funding agencies. The mention of commercial products, their source, or their use in connection with the material reported herein is not to be construed as an actual or implied endorsement of such products.

**Competing interests:** The authors have declared that no competing interests exist.

## Results

Seventy-seven students (age = 13.5±0.6 years; 50.4% female; median annual family income = $25K-$50K with 70.1% <$50K; 39 assigned to intervention group) were enrolled in the study. The teaching sessions did not result in any significant change in the participants' understanding domain scores in either the intervention or the control groups. However, the intervention (air pollution) teaching session resulted in a statistically significant increase in behavior domain score from baseline to immediately post-teaching, which continued to be present at one-month follow-up (mean ± standard deviation of score change immediately after = 1.7±3.3; score change 1-month after = 2.5±3.2; P<0.001; minimally important difference = 1.0).

## Discussion

This pilot study highlights the potential of a short one-time teaching session in promoting environmentally friendly behavior choices among adolescents.

## Introduction

Air pollution is a major environmental problem with important health consequences that affects billions of people around the world [1–4], and has been shown to be associated with increased morbidity and mortality [5, 6]. Air pollution has been linked to premature birth, impairment in cognition and lung development, cardiovascular disease, and cancer [7, 8]. Despite the wealth of scientific information on the health effects of air pollution, the public's personal choices in their daily lives (for example, use of public versus private transportation or use of locally produced versus imported food) continue to be largely detrimental towards the environment [9], and public support for generation of policies to regulate sources of pollution has remained modest at best.

One potential factor associated with the disconnect between the state of the scientific knowledge and the public's behavioral and lifestyle choices towards the environment may be due to a lack of awareness and understanding of the available knowledge [10, 11]. While there has been substantial media attention and increased publicity to the ongoing environmental problems in the recent years, the level of public enthusiasm and motivation to take the necessary steps to battle air pollution by modifying their behavior and lifestyle has remained low (for example, economic prosperity and reducing healthcare costs are considered of much higher priority as opposed to protecting environment and dealing with climate change) [12]. Interestingly, at least in the developed countries, such attitudes seem not to be necessarily governed by income levels [13]. An important impedance to the pursuing environmentally friendly behavioral choices may be the established lifestyle demands of the current generation of adults. The current generations of adults have mostly raised with little understanding of the human contribution to air pollution and their impact on the environment, making any substantial change to their behavior challenging [14–16]. A potential solution to this challenge may be to see whether providing appropriate education to the younger generation could result in lifestyle changes beneficial towards the environment since their lifestyles and habits are currently unformed and more amenable to influence. That said, it is currently unclear whether teaching adolescents about the health effects of air pollution would affect their future awareness of air pollution or result in more environmentally sound future behavior.

The aim of this study was to determine whether a short one-time interactive teaching session on air pollution can shift the behavioral choices of adolescents towards environmentally friendlier options. We hypothesized that educating adolescents about the health effects of air pollution through a short one-time interactive evidence-based teaching session would improve their long-term understanding of human contribution to air pollution and shape their behavior to make personal choices that are more environmentally friendly. To examine this hypothesis, we developed a one-hour interactive evidence-based teaching session about air pollution and its effects on climate change and human health along with a questionnaire survey with domains to assess understanding and behavioral choices. The teaching session and questionnaire were then used in a randomized controlled trial to examine the hypothesis stated above.

## Methods

### Study design

To examine the long-term efficacy of a short one-time teaching session about air pollution on the behavioral choices of adolescents concerning environmental health issues, we developed a one-hour interactive script-based evidence-supported teaching curriculum about air pollution and its effects on climate change and human health. To quantify the effect of the teaching session, we developed a survey questionnaire with two topic domains to measure the understanding and future behavioral choices of the participating adolescents towards air pollution. To determine the longitudinal effectiveness in promoting environmentally friendly behavior, we planned to administer the questionnaire immediately before, immediately after and then at intervals of three, six, and twelve months. To allow for scholastic holidays, the questionnaire was allowed to be administered within two months before or after the planned timelines (e.g., one to five months for three-month time point). Furthermore, to determine whether teaching about air pollution, versus the process of providing education by educators or researchers, is effective in promoting environmentally friendly behavior, we also developed a parallel "control" teaching session about the health benefits of vaccination and used it as the control teaching session. We then performed a pilot randomized controlled trial in which the 8[th] grade students were randomized to receive the 1-hour teaching session about air pollution's effects on lung health (intervention group) or a separate teaching session on the role of vaccination in public health (control group). The *a priori* primary outcome of the study was the questionnaire behavioral domain score. The secondary outcome was the questionnaire understanding domain score.

Full study protocol is available at dx.doi.org/10.17504/protocols.io.j8nlko9bxv5r/v1, and also available in part in Supplemental S1 File.

### Study population

The inclusion criteria for participation in the study included (1) being a middle school students aged 13 to 15 years of age; (2) ability to read, understand, and write in English at the middle school level; and (3) willingness to participate in the follow-up visits of the study. The exclusion criteria included (1) inability to complete the pre-intervention questionnaire; and (2) known plans to move out of the area within the next 6 months after enrollment in the study. As an initial attempt, teachers from the San Jose, California metropolitan area KIPP school (Knowledge Is Power Program), a national network of free open-enrollment college preparatory public schools dedicated towards students in underserved communities, were invited to participate in the study by collaborating with study recruitment and educational activities. Air pollution levels are consistently higher in the South San Francisco Bay and in the San Jose Metropolitan area compared to the neighboring parts. For example, the highest

average pollutant concentration for ozone (over an eight-hour period) and particulate matter of 10 micron or less (over a 24-hour period) were 81 ppb and 77 mcg/m$^3$, respectively, in San Jose in 2019. By comparison, those levels were 73 ppb and 42 mcg/m$^3$ in San Francisco [17].

Beginning February 1, 2015, students from all three 8th grade classrooms at the San Jose KIPP school were invited by their teachers to participate in the study. The teachers then distributed a letter of invitation along with assent and consent forms to the interested students to take home and review with their parents/guardians, and later collected and provided the forms to the study investigators. Students who provided signed consent and assent forms were included in the study if they met the enrollment criteria. The consent and ascent forms were provided in English and Spanish language to allow for language preference of the students' parents and/or guardians in this school with majority Hispanic population. However, the teaching sessions and the questionnaire administered to the students were only in English, as the KIPP students were considered to all be proficient in reading and writing English language at the level of middle school. The study enrollment ended on December 20, 2018.

## Trial regulatory matters

The University of California San Francisco (UCSF) Institutional Review Board (IRB) and the San Francisco Veterans Affairs Health Care System (SFVAHCS) Committee on Research and Development approved the study protocols. The study was registered with the United States (U.S.) National Library of Medicine (Behavioral Research of Environment and Air Pollution Through Education (BREATHE) study; ClinicalTrials.gov identifier NCT02471872). All participating adolescents and their parents or guardian provided IRB-approved written informed assent and consent, respectively, as well as Health Insurance Portability and Accountability Act (HIPAA). No monetary compensation was provided to the participants or their family.

## Recruitment

Recruitment was done through primary contact by KIPP school teachers. Teachers offered adolescent students in the 8th grade classrooms to participate in the study, and provided them with an IRB-approved study description along with assent and consent forms. The students who provided written assent and parental or guardian consent were enrolled in the study. Students who declined to participate were allowed to have a separate educational activity through their KIPP school teachers during the one-hour conduct of the study.

## Randomization

The participating students were assigned a study ID number through sorting their last names (and first names if needed) alphabetically and then assigning each student a study ID number. They were then partitioned by simple randomization of their subject ID number within their classrooms using STATA software to either the intervention (air pollution teaching) or control (vaccine teaching) group such that about equal numbers of students in each classroom were assigned to intervention and control groups. The study staff and students were not blinded to the intervention assignment; however, the staff scoring and collecting data from the questionnaires were blinded to the randomization assignment. The data was unblinded by the study statistician who was not involved with the administration or scoring of the questionnaires at the completion of the study.

## Teaching sessions

Because the main objective of this study was to determine whether educating adolescents about air pollution in the form of a single teaching session would result in behavioral changes that are environmentally friendly, we chose a one-hour in-person teaching session about air pollution by researchers as the intervention. For the control group, we chose to provide an in-person teaching session by researchers about another topic other than air pollution (vaccination). This allowed us to test the hypothesis that what the researchers taught (versus the researchers' presence) will affect the adolescents' future behavior.

The teaching sessions were one hour in duration, script-based, and associated with a PowerPoint slide presentation (https://arjomandilab.ucsf.edu/questionnaires-powerpoints). The intervention (air pollution) teaching session was developed by our research group, which comprised researchers and educators including an educator from the San Jose KIPP school, and based on the available scientific literature on air pollution and its health effects from the United States Environmental Protection Agency and World Health Organization [18, 19]. The session involved an interactive session that included undergoing a breathing test (spirometry) using a handheld spirometer (EasyOne spirometer; ndd Medical, Inc., Zurich, Switzerland). The Control (vaccination) session also involved an evidence-supported script-based session developed by our research group based on the information on vaccination and its benefits available from the United States Centers for Disease Control website and The Public Broadcasting Company [20, 21].

The participating students were separated into two groups according to their group assignments and placed into two different classrooms. The assigned teaching sessions were administered by the study staff to each group.

## Questionnaire

The questions on the questionnaire were formulated collaboratively by our research team that included researchers and educators (including an educator from the San Jose KIPP school). The questionnaire was specifically developed for this study on the basis of the hypothesis that providing knowledge that result in improved risk perception and outcome expectancy principles towards environmental health, as outlined in Social Cognitive Theory [22, 23], is best achieved in adolescent age group. The questions were carefully crafted, considering the age group of the target population (middle adolescents in 13 to 15 years of age) and the current environmental issues, and were designed to provide a quantification of perspective participants' level of understanding environmental health issues as well as degree of interest in making lifestyle choices that may be perceived as environmentally friendly. To do this, the questions were designed to assess several components, including knowledge of environmental pollution and its health effects, perception of how much participants think and discuss environmental issues, and attitudes towards daily activities and lifestyle choices aimed at mitigating air pollution and climate change.

The questionnaire was developed to assess two domains: (1) adolescents' understanding of air pollution, its sources, and its environmental and health effects and (2) adolescents' future behavior based on their report of personal behavioral choices as well as their support for public policies with potentially important effects on air pollution, environment, and climate change (https://arjomandilab.ucsf.edu/questionnaires-powerpoints). It consisted of 15 questions which were divided into the two domains gauged towards quantification of adolescents' understanding and behavioral choices. The questionnaire generated two scores, which were weighted sums of the questions in their perspective sections as described below. The multiple-choice answers to each question were weighted to have a minimum score of 1 and a maximum

score of 4. The understanding domain section contained 5 multiple-choice questions designed to target and quantify the understanding of the participants with a score ranging from a minimum of 5 to a maximum of 20 possible points. The behavior domain section contained 10 questions (7 multiple-choice and 3 free-narrative response) designed to target and quantify behavioral choices of participants with a score ranging from a minimum of 10 to a maximum of 38 possible points. The free-narrative responses were scored by two blinded observers based on an *a priori* guideline from no response or an irrelevant response to a relevant response or a response that was related to recent relevant events.

The questionnaire was administered to all participating students on three different occasions regardless of their assignment to intervention or control group including one administration immediately before participation in the teaching sessions and one immediately after participation in the teaching sessions. The questionnaire was also administered to all participating students about one month after participation in the teaching session by their classroom teacher during their regular classroom session.

The structural validity (reliability) of the questionnaire was evaluated by computing the correlation coefficient of the data collected before and immediately after the teaching sessions from all participants. To account for any potential effects from the intervention administered, the correlation coefficient of the data was also computed within the control group. The content validity was evaluated by factor analysis with "varimax" rotation using data collected from all participants before the teaching sessions. Barlett's test of sphericity was performed to test whether data was appropriate for factor analysis [24, 25]. The number of factors to extract from factor analysis was determined using Scree plot and parallel analysis [26, 27]. Factor loadings and percentage of variance explained by the factors were reported.

## Sample size and power calculation

For sample size and power calculation, we made the following assumptions: (1) minimally important difference in questionnaire score of 10% change due to the intervention, (2) standard deviation of change in score of 40%, and (3) a drop-out rate of about 20% due to factoring including movement of subjects out of the area. A sample size of 504 subjects (252 subjects in each group) will provide a power of 80% to detect a change in questionnaire score by t-test with a two-sided type I error of 0.05. Considering the drop-out rate, we proposed to recruit a total of 600 subjects (300 in each group).

## Data availability

The study dataset generated and analyzed for this report is available on Dryad [28].

## Data management and statistical analysis

Data from the questionnaire was entered into a database by two research assistants blinded to the assignment of the participants. Participants' characteristics were examined and summarized within all participants and with respect to intervention and control groups. A comparison of the distributions was performed using an unpaired t-test for each continuous variable or a Chi-squared test for each binary or categorical variable. The P-values and the descriptive statistics including the mean ± standard deviation (SD) for continuous variables or the number and percentage of participants out of the total number of participants [n (%)] for binary and categorical variables were presented.

The distribution of total scores, as well as scores from understanding and behavior domains of the questionnaire, were examined both within-participants and between intervention and control groups. The differences in the scores from before to immediate or 1-month after the

intervention within each participant were examined using paired t-test. Changes in scores within each participant from before to immediate or 1-month after the intervention were obtained by subtracting the baseline scores from the later scores, respectively. The changes in questionnaire scores (outcomes) within each group were examined using t-test comparisons. Regression analyses using generalized estimating equations (GEE) were also performed to examine the changes in outcomes with a repeated measure design and adjustment for covariates including age (interval), sex (binary), race (nominal), ethnicity (binary), income level (ordinal), and assigned classrooms (nominal) as appropriate. Given that only few students provided race classification (22 out of 77) and the skewed distribution of family income levels of participants, only age, sex, and ethnicity were included in the final regression modeling. Furthermore, linear regression modeling with adjustment for covariates was performed for comparison of changes between the intervention and the control group at different time points.

   Data management and statistical analyses were conducted in Stata/IC (version 14.2, Stata Corp LP, College Station, TX, USA) and R software (version 4.2.2; R Foundation for Statistical Computing, Vienna, Austria). Figures were generated in GraphPad software (Prism version 7.0, San Diego, CA, USA). A significance level of $\alpha < 0.05$ was used to determine statistical significance.

## Determination of minimally important difference of the questionnaire

To compute estimates of the minimally important difference (MID) of the questionnaire and its domains, we determined MID through various distribution-based approaches using standard error of mean (SEM) and standard deviation of mean (SD) for baseline measurements as well as effect size approaches, as previously utilized by other investigators for estimation of various patient-reported outcomes [29, 30]. In particular, we used the various suggested approaches including 1, 1.96, and 2.77 multiplications of SEM and 0.5 multiplication of SD of baseline measurement [30–33]. Effect size approaches included the change in score observed in the intervention group normalized by SD of baseline measurements (effect size approach) or SD of change from baseline measurement (standardized response mean or SRM) [34–37]. Furthermore, reliable change index (RCI), defined as the change in score observed in the intervention group normalized by square root of SEM of change, was also calculated with an RCI of ≥1.96 considered to signify a clinically important change [29, 38].

## Results

### Participants' characteristics

The study enrollment began on February 1, 2015 and students from three classrooms at San Jose KIPP school constituted the first wave of enrollment. Although the study remained open to enrollment until December 20, 2018, no further students were enrolled in the study due to funding limitations. The study exclusively enrolled students from a single school within the span of one academic year. These enrolled students completed the same survey solely immediately before, immediately after, and at about one month (36 to 47 days after the teaching session).

   Altogether, a total of 105 students and their families were initially approached for participation in the study. Out of these, 25 declined assent and/or consent. Of the 80 students who provided assent and parental consent, one withdrew consent, and two declined to continue with the study on the day of the teaching session. Altogether, 77 students were enrolled and randomized in the study, out of whom 39 (51%) were randomized to the intervention group and 38 (49%) to the control group (Fig 1).

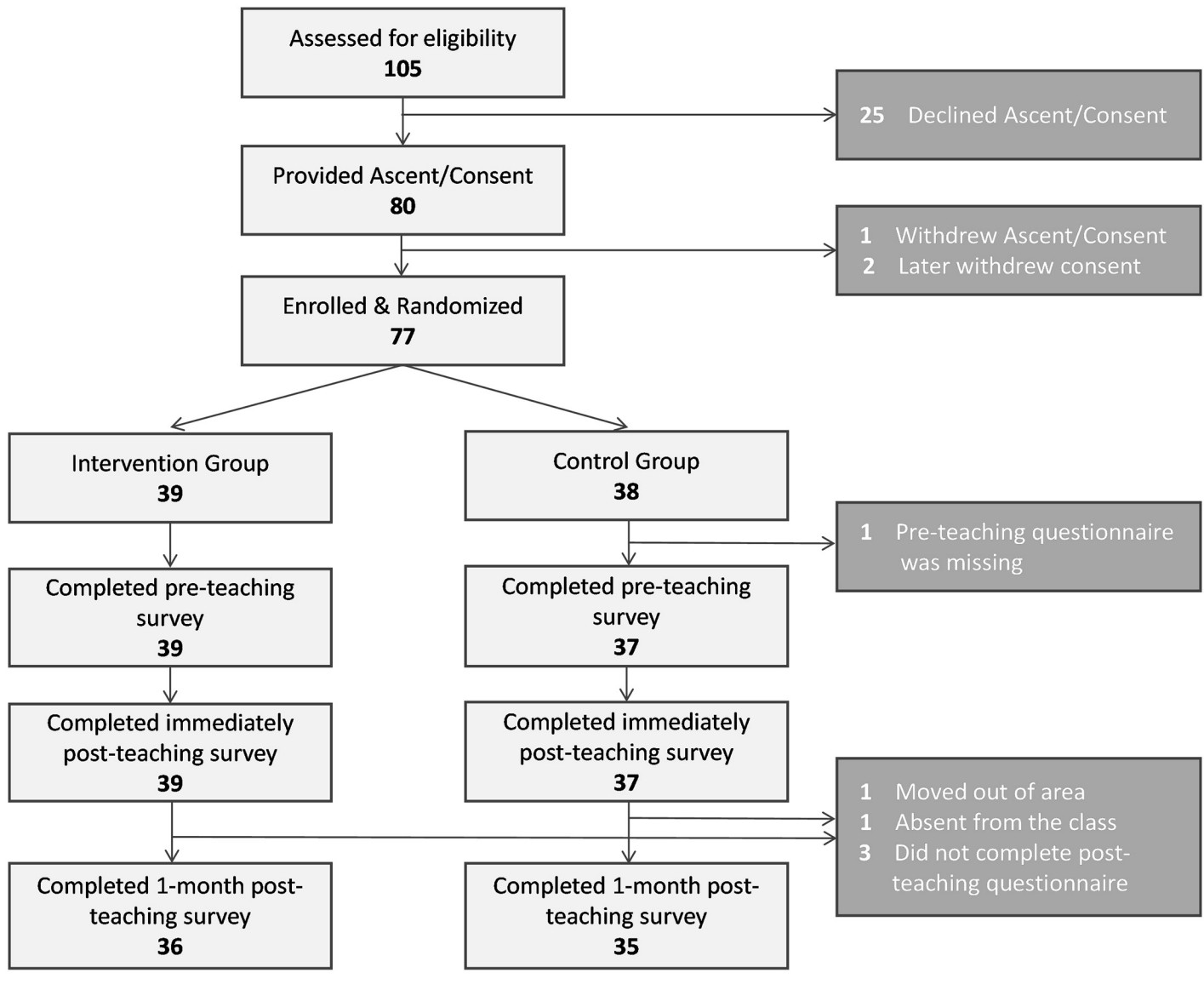

**Fig 1. Study design.**

From these 77 students, pre- or post-teaching session questionnaires were unavailable from 6 students: the pre-teaching session questionnaire was missing from one student; during the one-month follow-up survey, two students were absent from the classrooms (one was absent for an unknown reason; another moved out of the area and was no longer a student at the school) and three students did not complete the questionnaire (reasons unknown). Overall, complete data for analysis was available from 36 students in the intervention group and 35 students in the control group.

The characteristics of the participating students who were enrolled and randomized are shown in Table 1. The participants had a mean age of 13.5 ± 0.6 years and were approximately half male and half female. The intervention and control groups were comparable with regards to age and gender distributions. Most participants were of Hispanic ethnicity (82%). Most of those who identified their ethnicity as Hispanic did not provide a race category. There was no

**Table 1. Participants characteristics.**

| | All | Intervention | Control |
|---|---|---|---|
| **No. of participating students** | 77 | 39 | 38 |
| **Demographics** | | | |
| **Age in years** (Mean ± SD) | 13.5 ± 0.6 | 13.4 ± 0.6 | 13.5 ± 0.5 |
| **Male sex** [n (%)] | 38 (49.4) | 19.0 (48.7) | 19 (50) |
| **Ethnicity** | | | |
| Not Hispanic [n (%)] | 9 (11.7) | 6 (15.4) | 3 (7.9) |
| Hispanic [n (%)] | 63 (81.8) | 31 (79.5) | 32 (84.2) |
| Unknown [n (%)] | 5 (6.5) | 2 (5.1) | 3 (7.9) |
| **Race** | | | |
| Native American or Alaskan Native [n (%)] | 0 (0) | 0 (0) | 0 (0) |
| Asian [n (%)] | 8 (10.4) | 5 (12.8) | 3 (7.9) |
| Black or African American [n (%)] | 4 (5.2) | 3 (7.7) | 1 (2.6) |
| Native Hawaiian [n (%)] | 0 (0) | 0 (0) | 0 (0) |
| Other Pacific Islander [n (%)] | 2 (2.6) | 2 (5.1) | 0 (0) |
| White [n (%)] | 6 (7.8) | 3 (7.7) | 3 (7.9) |
| Mixed race [n (%)] | 2 (2.6) | 1 (2.6) | 1 (2.6) |
| Unknown [n (%)] | 55 (71.4) | 25 (64.1) | 30 (78.9) |
| **Income level** | | | |
| Less than $25K [n (%)] | 23 (29.9) | 15 (38.5) | 8 (21.1) |
| $25K-$50K [n (%)] | 31 (40.3) | 15 (38.5) | 16 (42.1) |
| $50K-$75K [n (%)] | 10 (13.0) | 5 (12.8) | 5 (13.2) |
| $75K-$90K [n (%)] | 4 (5.2) | 1 (2.6) | 3 (7.9) |
| More than $90K [n (%)] | 5 (6.5) | 1 (2.6) | 4 (10.5) |
| Unknown [n (%)] | 4 (5.2) | 2 (5.1) | 2 (5.3) |

Summary statistics for the characteristics of the participants are shown. Data are presented as mean ± standard deviation or number and percentage of participants, n (%), out of the total number of participants. Information about income levels was obtained using categories based on the US median income and poverty levels [14].

statistically significant difference in race or ethnicity categories between the two groups. Compared to the control group, a statistically significantly higher proportion of participants in the intervention group reported a family income of less than $25,000 per year (21% in the intervention group compared to 39% in the control group; P = 0.036).

## Questionnaire performance

Data for questionnaire performance was summarized in Table 2. At baseline, the questionnaire total scores ranged from 15 to 49 (total possible range was 15 to 58) with the understanding domain ranging from 5 to 20 (total possible range was 5 to 20) and the behavior domain ranging from 10 to 29 (total possible range was 10 to 38). Immediately after teaching sessions, the questionnaire total scores ranged from 24 to 47 with the understanding domain ranging from 8 to 20 and the behavior domain ranging from 11 to 30. At one month follow-up visit, the questionnaire total scores ranged from 21 to 49 with the understanding domain ranging from 9 to 20 and the behavior domain ranging from 11 to 32 (Fig 2).

The test-retest reliability of the questionnaire computed using data collected from all participants before and immediately after the teaching sessions was 0.77. The test-retest reliability was 0.75 when it was computed using data from the control group only. Barlett's test of sphericity showed a significant P value <0.001and thus granted appropriateness of factor analysis.

**Table 2. Questionnaire scores.**

|  | All | Intervention | Control |
|---|---|---|---|
| **Baseline** |  |  |  |
| Total | 36.2 ± 5.7 | 36.3 ± 6.2 | 36.0 ± 5.2 |
|  | 37.0 [31.0, 41.0] | 37.0 [32.0, 41.0] | 36.0 [31.0, 40.0] |
|  | {21.0, 49.0} | {21.0, 47.0} | {28.0, 49.0} |
| Understanding | 16.3 ± 3.2 | 16.5 ± 3.4 | 16.1 ± 2.9 |
|  | 17.0 [14.0, 19.0] | 17.0 [15.0, 19.0] | 17.0 [14.0, 18.0] |
|  | {5.00, 20.0} | {5.00, 20.0} | {9.00, 20.0} |
| Behavior | 19.9 ± 4.3 | 19.8 ± 4.5 | 19.9 ± 4.1 |
|  | 21.0 [17.0, 23.0] | 21.0 [16.5, 24.0] | 21.0 [17.0, 22.0] |
|  | {10.0, 29.0} | {10.0, 27.0} | {11.0, 29.0} |
| **Immediately after** |  |  |  |
| Total | 37.5 ± 5.5 | 38.2 ± 5.7 | 36.8 ± 5.3 |
|  | 37.0 [34.0, 42.0] | 37.0 [34.0, 43.5] | 37.5 [33.0, 40.0] |
|  | {24.0, 47.0} | {24.0, 47.0} | {25.0, 46.0} |
| Understanding | 16.5 ± 2.9 | 16.6 ± 3.1 | 16.3 ± 2.8 |
|  | 17.0 [14.0, 19.0] | 17.0 [15.0, 19.0] | 17.0 [14.0, 19.0] |
|  | {8.00, 20.0} | {8.00, 20.0} | {10.0, 20.0} |
| Behavior | 21.0 ± 4.2 | 21.6 ± 4.1 | 20.5 ± 4.2 |
|  | 22.0 [18.0, 24.0] | 22.0 [18.0, 25.0] | 21.0 [19.0, 23.0] |
|  | {11.0, 30.0} | {14.0, 29.0} | {11.0, 30.0} |
| **1 month after** |  |  |  |
| Total | 38.4 ± 6.4 | 38.4 ± 6.7 | 38.5 ± 6.1 |
|  | 39.0 [33.8, 43.0] | 40.5 [32.8, 43.3] | 38.0 [34.0, 42.0] |
|  | {26.0, 52.0} | {26.0, 49.0} | {29.0, 52.0} |
| Understanding | 16.1 ± 3.1 | 15.9 ± 3.2 | 16.4 ± 3.1 |
|  | 17.0 [14.0, 19.0] | 17.0 [15.0, 18.0] | 17.0 [14.0, 19.3] |
|  | {9.00, 20.0} | {9.00, 20.0} | {10.0, 20.0} |
| Behavior | 23.3 ± 4.8 | 22.4 ± 4.7 | 22.1 ± 5.0 |
|  | 22.0 [19.0, 26.0] | 23.0 [19.8, 26.3] | 22.0 [19.0, 26.0] |
|  | {11.0, 32.0} | {12.0, 29.0} | {11.0, 32.0} |

Summary statistics for the questionnaire scores are shown. Data are presented as mean ± standard deviation, median [1st quartile, 3rd quartile], and {minimum, maximum}.

Scree plot and parallel analysis of data suggested factor analysis with two factors. This analysis showed that using two factors separated the survey questions in accordance with the understanding and future behavior domains but explained only 19.4% of the variance (S1 Table in S2 File). We subsequently performed factor analyses for three, four, and five factors. With five factors, the percentage of variance explained by the factors improved to be 43.1% with survey questions quantifying behaviors loaded mostly on Factors 1, 3 and 4; while questions quantifying understanding loaded mainly on Factor 2 (S2 Table in S2 File).

The MID of the questionnaire was computed using distribution-based methods as shown in Table 3. Using SEM of baseline measurements, the MID was estimated to be between 0.5 to 1.8 for total score, 0.3 to 1.0 for understanding, and 0.4 to 1.4 for behavior. Using SD approach, the MID estimates were calculated to be higher in 2.0 range. The effect size approach generated a smaller estimate of MID in 0.5 or smaller range. Ultimately, a value of 1.0 was used as the most appropriate estimate of MID for this questionnaire total score and its domains.

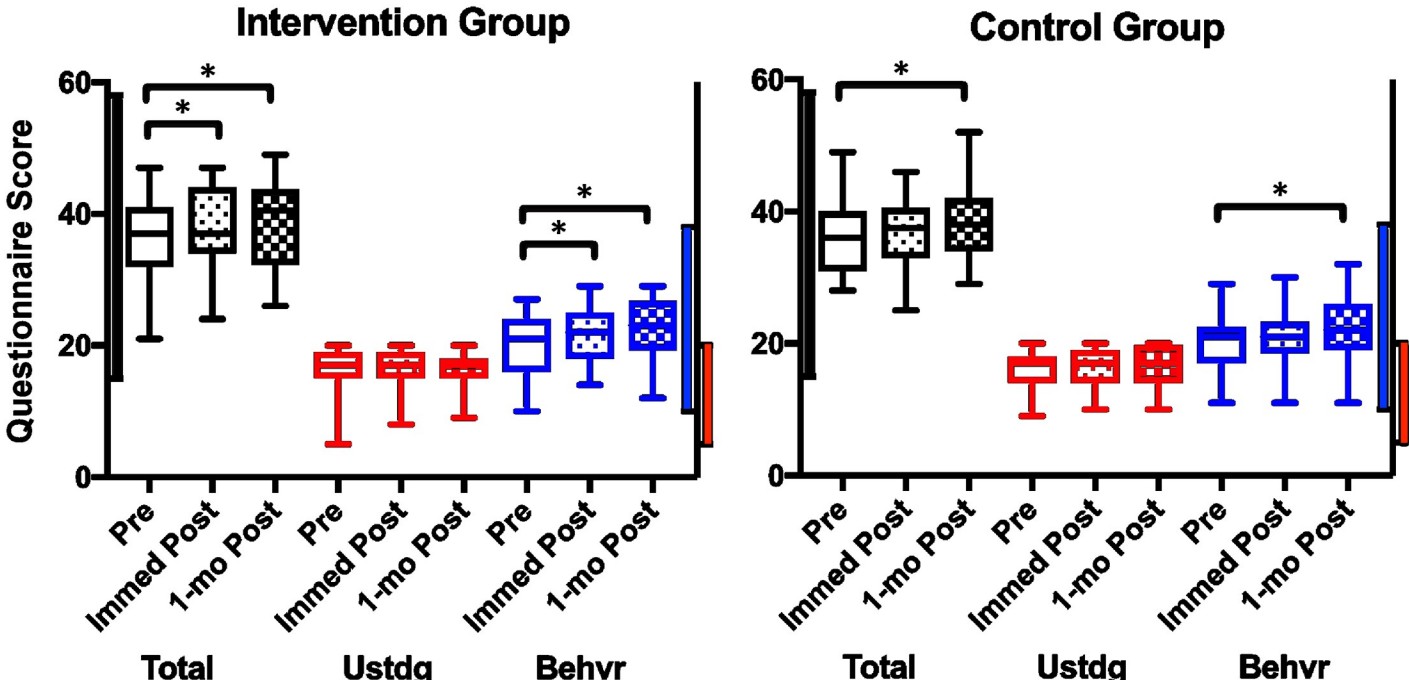

**Fig 2. Questionnaire scores at baseline and 1 month after the teaching session.** Boxplots of the scores from total (black), understanding (red), and behavior (blue) domains of the questionnaire assessed immediately before (Pre), immediately after (Immed Post), and 1-month after (1-mo Post) the teaching session for the intervention and control groups are shown. The scores were compared using paired t-test. Boxplots represent median and interquartile range; whiskers represent total range. The vertical bars on the y-axis represent the minimum and maximum scores possible for total (black), understanding (red), and behavior (blue) domains. Abbreviations: Ustdg = understanding; Behvr = behavior; * = paired t-test comparison P value <0.05.

**Changes in understanding and behavior measures from pre- to post-teaching.** Similar results were generated when the changes in questionnaire scores within each group were analyzed using paired t-test or multivariable regression (GEE) (GEE results presented in S3 Table in S2 File). The teaching sessions did not result in any significant change in the participants' understanding domain scores in either the intervention or the control groups (Fig 2, red box plots and Table 4). However, the intervention (air pollution) teaching session did result in a statistically significant increase in behavior domain score from baseline to immediately post-teaching, which was greater than the presumed MID for the questionnaire. Of note, this improvement continued to be present at 1-month follow up (Fig 2, panel A, blue box plots and Table 4). The control (vaccine) teaching session did not cause any significant change in the behavior domain score immediately post-teaching (Fig 2, panel B, blue box plots and Table 4).

The changes in questionnaire scores between the intervention and the control groups were analyzed using multivariable linear regression analyses. These analyses showed that the changes in behavior scores were statistically significantly higher in the intervention group compared to the control group (a differentially larger change [95% CI] of 1.7 [0.1, 3.3] in the intervention group; P = 0.037) at immediately post-teaching (Fig 3 and Table 5).

Interestingly, at 1-month follow-up survey, there was a significant increase from baseline in the behavior domain score of the control group (Fig 2, panel A, blue box plots and Table 4) such that although the increase in the intervention group behavior score was slightly higher than that of the control group, the difference was no longer statistically significant (a differentially larger change [95% CI] of 0.4 [-2.3, 1.4] in the intervention group, P = 0.637) (Fig 3 and

**Table 3. Determination of minimally important difference of the questionnaire.**

| Minimally Important Difference | Total Score | Understanding Score | Behavior Score |
|---|---|---|---|
| **Distribution Approaches** | | | |
| **Baseline measurements** | | | |
| **SEM of baseline measurements** | | | |
| 1 SEM | 0.65 | 0.36 | 0.49 |
| 1.96 SEM | 1.28 | 0.71 | 0.96 |
| 2.77 SEM | 1.80 | 1.00 | 1.35 |
| **SD of baseline measurements** | | | |
| 0.5 SD | 2.84 | 1.58 | 2.13 |
| **Intervention group measurements** | | | |
| **Effect Size** | | | |
| 0.2 effect size | 0.07 | -0.03 | 0.12 |
| 0.5 effect size | 0.18 | -0.07 | 0.29 |
| 0.8 effect size | 0.29 | -0.11 | 0.46 |
| **Standardized response mean of effect size** | | | |
| 0.2 effect size | 0.08 | -0.03 | 0.13 |
| 0.5 effect size | 0.21 | -0.07 | 0.32 |
| 0.8 effect size | 0.33 | -0.12 | 0.51 |
| **Reliable change index (RCI)** | | | |
| Index | 2.40 | -0.62 | 3.39 |
| >1.96 | Yes | No | Yes |

Minimally important difference was assessed through various distribution-based approaches using standard error of mean (SEM) and standard deviation of mean (SD) for baseline measurements, effect sizes as well as the reliable change index (RCI), defined as the change in score observed in the intervention group normalized by square root of SEM of change. Abbreviations: SEM = standard error of mean; SD = standard deviation; RCI = reliable change index.

**Table 4. Change in questionnaire scores immediately and one-month after the teaching session intervention.**

| Questionnaire Responses | | Immediately after | | One-month after | |
|---|---|---|---|---|---|
| | No. | Mean (95% CI) | P value | Mean (95% CI) | P value |
| **Total Score** | | | | | |
| All participants | 77 | **1.4 (0.4, 2.3)** | **0.004** | **2.2 (1.0, 3.3)** | **<0.001** |
| Intervention | 39 | **1.9 (0.7, 3.2)** | **0.004** | **2.0 (0.6, 3.5)** | **0.007** |
| Control | 38 | 0.8 (-0.6, 2.1) | 0.259 | **2.3 (0.4, 4.2)** | **0.019** |
| **Understanding Score** | | | | | |
| All participants | 77 | 0.2 (-0.4, 0.7) | 0.545 | -0.1 (-0.8, 0.6) | 0.816 |
| Intervention | 39 | 0.2 (-0.6, 1.0) | 0.640 | -0.4 (-1.5, 0.6) | 0.400 |
| Control | 38 | 0.2 (-0.7, 1.0) | 0.702 | 0.3 (-0.7, 1.3) | 0.573 |
| **Behavior Score** | | | | | |
| All participants | 77 | **1.2 (0.4, 1.9)** | **0.001** | **2.3 (1.3, 3.2)** | **<0.001** |
| Intervention | 39 | **1.7 (0.7, 2.8)** | **0.002** | **2.5 (1.4, 3.6)** | **<0.001** |
| Control | 38 | 0.6 (-0.4, 1.6) | 0.237 | **2.0 (0.5, 3.6)** | **0.013** |

The change in total, understanding domain, and behavioral domain scores of participants in intervention and control groups from prior to attending the teaching session immediately after and one month after the teaching session are shown. The number of participants for each comparison (No.), mean change of the scores with 95% confidence interval (CI), and P values from paired t-test comparing pre- and post-teaching session scores are shown. Statistically significant changes and the corresponding P values were shown in bold.

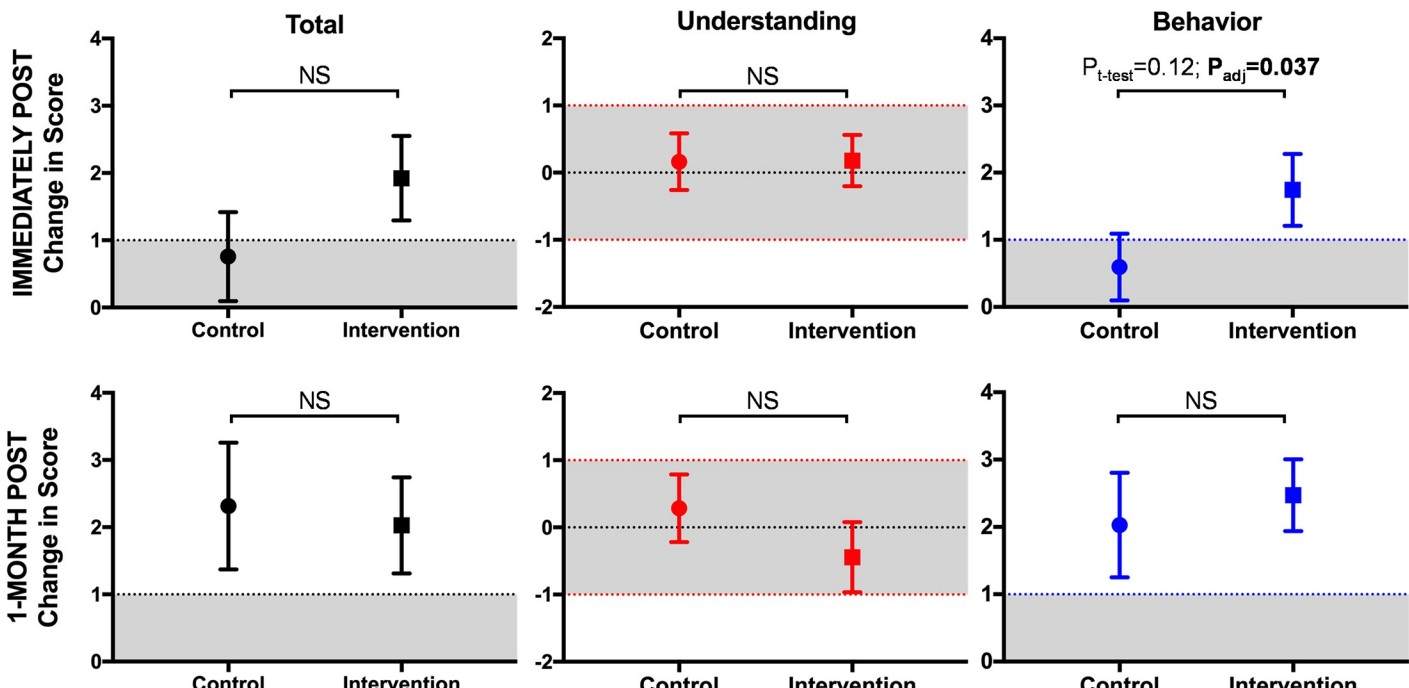

**Fig 3. Change in questionnaire scores after the teaching session.** The graph shows mean and standard error of mean of change in the score from total (black), understanding (red), and behavior (blue) domains of the questionnaire at immediately and one month after the teaching intervention. The dotted lines show minimally important difference (MID) limits and the gray areas show changes that are less than MID. P values from unpaired t-test ($P_{t\text{-test}}$) and regression with adjustment for age, sex, and ethnicity ($P_{adj}$) are shown. Abbreviations: NS = not significant from unpaired t-test comparison; MID = minimally important difference.

Table 5). The total score values and changes mirrored those of behavior domain (Fig 2, black box plots and Fig 3).

At 1-month follow-up, the changes in the total and the behavior domain scores in both intervention and control groups exceeded the presumed estimated MID of 1.0. The change in the understanding domain score did not exceed the MID in either group (Fig 3).

## Discussion

We developed a one-hour teaching session about air pollution and climate change targeted towards adolescents along with a questionnaire to quantitatively assess adolescents' understanding and behavioral choices regarding air pollution and environment. We then conducted a randomized controlled trial using the teaching session and the questionnaire to see whether this one-time short teaching session about air pollution and climate change, versus the control session about vaccines, could have lasting effects on understanding of air pollution and climate change issues and behavior of adolescents in choosing environmentally friendlier approaches in their daily living. We found that our air pollution teaching session did not affect how the adolescents scored on the questionnaire understanding domain. However, we found that the air pollution teaching session did improve the behavioral choices of the adolescents as measured by the questionnaire, both immediately and about one month after the administration of the teaching session. The changes in the behavioral domain score were larger than the estimated MID of the questionnaire, suggesting a statistical and clinically significant improvement lasting over a month with a single one-hour teaching session about environment and health. To our knowledge, this is the first study to show that a one-time teaching session on air

**Table 5. Comparison of questionnaire scores between intervention and control groups.**

| Questionnaire Responses | Difference (95% CI) | P value (t-test) | PE (95% CI) | P value (adjusted regression) |
|---|---|---|---|---|
| **Pre-intervention scores** | | | | |
| Total | 0.3 (-2.9, 2.3) | 0.846 | -1.0 (-3.7, 1.6) | 0.443 |
| Understanding | 0.4 (-1.8, 1.1) | 0.629 | -0.6 (-2.1, 1.0) | 0.460 |
| Behavior | -0.1 (-1.9, 2.1) | 0.920 | -0.5 (-2.5, 1.6) | 0.646 |
| **Immediate post-intervention scores** | | | | |
| Total | 1.4 (-3.9, 1.1) | 0.264 | 1.0 (-1.6, 3.6) | 0.437 |
| Understanding | 0.4 (-1.7, 1.0) | 0.602 | -0.3 (-1.7, 1.1) | 0.655 |
| Behavior | 1.1 (-3.0, 0.8) | 0.265 | 1.3 (-0.5, 3.2) | 0.159 |
| **Change in scores from baseline** | | | | |
| Total | 1.2 (-3.0, 0.6) | 0.204 | 1.8 (-0.1, 3.8) | 0.066 |
| Understanding | 0.02 (-1.2,1.1) | 0.975 | -0.2 (-1.1, 1.4) | 0.799 |
| Behavior | 1.1 (-2.6, 0.3) | 0.120 | **1.7 (0.1, 3.3)** | **0.037** |
| **1-month post-intervention scores** | | | | |
| Total | -0.1 (-2.9, 3.1) | 0.941 | -0.8 (-3.9, 2.2) | 0.590 |
| Understanding | -0.4 (-1.0, 1.9) | 0.552 | -1.2 (-2.6, 0.3) | 0.125 |
| Behavior | 0.3 (-2.6, 1.9) | 0.771 | 0.3 (-2.0, 2.7) | 0.786 |
| **Change in scores from baseline** | | | | |
| Total | -0.3 (-2.1, 2.7) | 0.808 | 0.3 (-2.2, 2.8) | 0.821 |
| Understanding | -0.7 (-0.7, 2.2) | 0.317 | -0.7 (-2.3, 0.9) | 0.389 |
| Behavior | 0.4 (-2.3, 1.4) | 0.637 | 1.0 (-1.0, 3.0) | 0.333 |

The comparison of total, understanding domain, and behavioral domain scores and changes in those scores between participants in the intervention and the control groups over the three assessments (pre-, immediate post-, and one-month post-intervention) are shown. Numbers represent mean difference and the parameter estimate (PE) from unpaired t-test and regression modelling adjusted for age, sex, and ethnicity, respectively, with corresponding 95% confidence interval (CI) and P value.

pollution pollutants, climate change, and environmental health in adolescents could cause a shift in their choices towards environmentally friendlier behavior.

There are few studies that have examined the benefits of environmental education in youth, but most have been descriptive and cross-sectional in nature, looking at the association of environmental knowledge and environmentally friendly behavior. In 2011, Naquin and colleagues designed a questionnaire to investigate environmental health knowledge, attitudes, and practices of 115 children enrolled in grades four through eight at a university laboratory school in southeast Louisiana, and reported the differences in those domains by students' grade levels [39]. Later in 2023, Elshaer and colleagues reported developing a questionnaire that was fashioned after that of Naquin et al. to assess the environmental knowledge, attitudes, and behaviors of 452 children ranging from 9 to 18-years-old from three schools in the Cincinnati, Ohio area. In their cross-sectional study, Elshaer and colleagues did not find any association between environmental knowledge and behavior, and concluded that health knowledge does not necessarily translate to action in youth [40]. In 2022, Bøhlerengen and colleagues also designed a questionnaire and evaluated the environmental attitudes, behaviors, and responsibility perceptions among 211 Norwegian youth between the ages of 16 to 20 years and reported no significant correlation between the youth's knowledge of air pollution dangers and the various factors in a behavioral developmental model examined. They concluded that environmental education for youth should not only focus on increasing knowledge, but also on promoting positive development to improve their contribution towards environmental health [41]. These

reports are consistent with our observation that despite the lack of any significant change in the understanding domain score with the environmental teaching session, there was an improvement in behavioral choice domain score. Two possible explanations for lack of improvement in the understanding scores in our study may be offered: (1) the teaching session was not as informative for the education level of the adolescents targeted, and/or (2) the understanding domain questions of the survey were too basic and did not have adequate resolution to detect a change in the understanding of the students. The distribution and spread of the understanding score at the baseline with several students achieving the highest possible score on the questionnaire are supportive of the latter possibility, although the presence of other students who scored at the bottom of the range would argue against such a hypothesis. By contrast, the behavior domain score distribution was more centered in the middle of the possible score, thus allowing more improvement in that domain. Overall, the change in the behavior domain score and the lack of change in the understanding domain score suggests that although the teaching session may not provide any additional educational value for the adolescents, it may still result in improvement in future behavioral choices, suggesting that simple reminders could have positive long-term effects on behavior.

A few other studies have reported examining the effect of environmental education in form of an intervention and examining its effects on youth; however, these studies were usually of a much longer duration teaching, were administered to older youth in the high school through college age range, were not evaluating specific environmentally-relevant behavioral changes, and/or lacked appropriate control arm [42–46]. For example, a 2019 study of 578 secondary school students in the 15 to 16 years of age range in Southeast Asia (Laos or Lao People's Democratic Republic) has documented that environmental education over a semester (3 sessions over 2 weeks as part of a longer health education curriculum of 24 sessions over 8 weeks) can lead to changes in risk perception and information-seeking behaviors towards air pollution in the intervention group. However, this study included a longer term educational activity and their survey lacked any specific questions for assessment of changes in relevant environmentally friendly behavior [42]. Another study, conducted between 2000 and 2001 at Dalhousie University, Canada, delved into the behavioral effects of environmental education in college students. Its focus was on gauging the impact of an environmental studies course, which included every other week seminars and twice-weekly lectures during two semesters. To measure the effect, the researchers administered a pre-existing questionnaire developed by Kempton et al. (1995) [47], both before and after the course duration. Notably, the findings showed a noticeable improvement in environmentally responsible behavior after the conclusion of the course. Nevertheless, this was a longer educational program, with no long-term follow up beyond the conclusion of the course, and lacked any control arm [43].

Interestingly, in our study, both adolescents who received the intervention (air pollution) teaching session and those who received the control (vaccine) teaching session showed a statistically significant improvement on the questionnaire in their behavior domain scores at about one-month follow-up, with the intervention group having a larger, though not any longer statistically significant, increase in their score. The finding that even the control group that had a teaching session unrelated to air pollution or climate change (vaccine) also showed an improvement in their one-month follow-up behavioral choice score towards environmentally friendlier options is surprising but could be potentially explained as a case of a "spillover effect" [48], as the students were not blinded to the interventions. The knowledge of the air pollution-related experiment and the likely communication between the students assigned to the intervention (air pollution) teaching and those assigned to the control (vaccine) teaching may have resulted in extension of the air pollution teaching benefit beyond the direct recipients of the intervention [48]. Of note, our randomization approach in this study was not cluster-based

and similar number of students within the same class were assigned to receive the intervention or control teaching. A cluster-based approach with all students within each class receiving the same intervention could have possibly helped to avoid a spillover effect if larger number of classrooms were included.

## Limitations

Our study has several limitations. Custom-designed surveys could be prone to various biases in evaluation of the outcomes of interest, including but not limited to sampling, non-response, and survivorship biases among others [49–51]. Our pilot study is not exempt from some of these potential biases. For instance, the study had a small sample size and it was conducted within a single school with a relatively narrow demographic representation. Furthermore, the non-response rate was relatively high (near 25%). Moreover, the longitudinal follow-up was confined to one long-term assessment timepoint after the initial visit assessment. Nevertheless, the uniform demographics and socioeconomic status of the student population, coupled with the minimum loss to follow-up likely counter the concerns for non-response and survivorship biases. Furthermore, biases can potentially arise from the specific metrics used to score such surveys, such as question order, answer arrangement, and conformity biases. However, the orders of the questions and answers on the survey had been in fact mixed, mainly to prevent any possibility of the participants copying answers from each other, but it also served to alleviate concerns associated with any biases stemming from question or answer order. Lastly, open-ended questions were incorporated, along with their specific scoring methodology, to reduce the effect of any conformity and social desirability bias.

Other limitations included the following. A particular shortcoming within behavior change studies is the sustainability of the intervention. Due to pilot nature of this study, a longer-term follow-up was not possible. Further research, particularly longitudinal studies utilizing a web-based questionnaire, could assess the sustainability of behavior change. Furthermore, the study population was not representative of the general population, as most students in our study were Hispanic. Future research with expanding study participants to include a more diverse group of students should help with understanding any contributory race and ethnicity effects. Finally, there was a limited span of the income level among the population studied with about two thirds of the participants reporting annual income levels below $50,000 (at or near poverty level in the US) [52]. This limitation may adversely affect the generalizability of the study findings, as the income level could potentially influence people's behavior towards environment [13]. However, our study was not powered to examine the contribution of socioeconomic factors in efficacy of the intervention administered. Future research could explore the role of socioeconomic factors in the effectiveness of educational interventions promoting pro-environmental behavior in adolescents.

## Conclusion

In conclusion, this study highlights the potential of a short one-time teaching session in promoting environmentally friendly behavior choices among adolescents. The findings of this study can inform the development of future interventions aimed at promoting environmental awareness and behavior change among adolescents and ultimately contribute to public and political support for aggressive regulation to protect the environment that could then result in reduction of the global public health burden of air pollution. Further research is necessary to fully understand the mechanisms behind these changes in behavioral choices, to determine the most effective ways to promote sustainable behavior among young people, and to explore the

long-term sustainability of these behavior changes and the factors that may influence their effectiveness.

## Supporting information

**S1 File. Study scientific protocol.**
(DOCX)

**S2 File. Supplemental tables.** S1 Table presented the factor loadings from factor analysis with two factors for the 15 questions of the questionnaire. S2 Table presented the factor loadings from factor analysis with five factors for the 15 questions of the questionnaire. S3 Table presented the results from multivariable regression analyses using generalized estimating equations with repeated measure design.
(DOCX)

## Acknowledgments

The authors would like to thank the teachers, staff, and the eighth-grade students and their families from the Knowledge Is Power Program (KIPP) San Jose, California metropolitan area for their participation in this research study. None of the participants or their family or the school staff or employees received any financial compensation through this study.

## Author Contributions

**Conceptualization:** Mehrdad Arjomandi.

**Data curation:** Siyang Zeng, Caroline Davidson, Emily Yao, Sarah Dalton, Mehrdad Arjomandi.

**Formal analysis:** Siyang Zeng, Mehrdad Arjomandi.

**Funding acquisition:** Siyang Zeng, Rachel Tenney, Mehrdad Arjomandi.

**Investigation:** Rachel Tenney, Caroline Davidson, Emily Yao, Sarah Dalton, Mehrdad Arjomandi.

**Methodology:** Siyang Zeng, Caroline Davidson, Emily Yao, Sarah Dalton, Mehrdad Arjomandi.

**Project administration:** Rachel Tenney, Caroline Davidson, Emily Yao, Sarah Dalton, Mehrdad Arjomandi.

**Resources:** Siyang Zeng, Mehrdad Arjomandi.

**Software:** Mehrdad Arjomandi.

**Supervision:** Rachel Tenney, Sarah Dalton, Mehrdad Arjomandi.

**Writing – original draft:** Yorusaliem Abrham, Siyang Zeng, Chantal Kloth, Mehrdad Arjomandi.

**Writing – review & editing:** Yorusaliem Abrham, Siyang Zeng, Rachel Tenney, Caroline Davidson, Emily Yao, Chantal Kloth, Sarah Dalton, Mehrdad Arjomandi.

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
