## [Decision Letter · Decision Letter 0]

19 Jun 2023

PONE-D-23-13463Effect of a single one-hour teaching session about environmental pollutants and climate change on the understanding and behavioral choices of adolescents: the BREATHE randomized controlled trialPLOS ONE

Dear Dr. Arjomandi,

Thank you for submitting your manuscript to PLOS ONE. After careful consideration, we feel that it has merit but does not fully meet PLOS ONE’s publication criteria as it currently stands. Therefore, we invite you to submit a revised version of the manuscript that addresses the points raised during the review process.

We look forward to receiving your revised manuscript.

Kind regards,

Timothy J Wade, Ph.D

Academic Editor

PLOS ONE

Reviewers' comments:

Reviewer's Responses to Questions

**Comments to the Author**

1. Is the manuscript technically sound, and do the data support the conclusions?

Reviewer #1: Partly

Reviewer #2: No

Reviewer #3: Partly

2. Has the statistical analysis been performed appropriately and rigorously? 

Reviewer #1: No

Reviewer #2: Yes

Reviewer #3: No

3. Have the authors made all data underlying the findings in their manuscript fully available?

Reviewer #1: Yes

Reviewer #2: Yes

Reviewer #3: No

4. Is the manuscript presented in an intelligible fashion and written in standard English?

Reviewer #1: Yes

Reviewer #2: Yes

Reviewer #3: Yes

5. Review Comments to the Author

Reviewer #1: The manuscript could be improved based on the comments below.

Detailed inclusion and exclusion criteria, detailed information on the questionnaires such as validation, reliability, the interpretation of the scores, and the sample size calculation are to be provided.

Based on the CONSORT statement, all statistical tests for baseline comparison should be avoided.

Page 17 Questionnaire performance and based on Figure 2, the median/IQR scores is to be provided.

Detailed regression analysis output to be presented in table form.

Page 18, the statement P=0.036 by regression analysis with adjustment for age, sex, and ethnicity)’ is to be separated and presented in another section of the results.

Table 3, 95% CI is to be included.

The statement ‘the limitation of our sample size’ is to be justified with sample size calculation.

In the study protocol, a number of other independent variables and potential confounding variables/covariates, and different statistical approach was mentioned. The differences are to be indicated and whether those variables were considered in this study and the analysis.

The citation in the text did not comply with the journal format.

Reviewer #2: Environmental education is essential for effectively communicating the harmful effects of air pollution and potential behavioral changes to reduce the risk of those impacts. While an important activity I do not think that the study design described by the authors is the one the test out their hypotheses. There are also a lot of missing details on how the authors developed the questionnaire and how they designed their teaching sessions.

Abstract

• I am a bit confused by the control group. I one hour teaching session was given to each group of different topics. I am not sure if this shows that the 1 hour teaching is effective of it has to do with the subject matter. I would think that the control group would be reading material of about air pollution and compare that to the teaching session.

Introduction

• The authors seem to be conflating two different but related issues: support for air pollution reduction policies/regulations and behavior changes. Given the premise of the study I recommend focusing on the behavioral changes and possible mention how that can lead to increased awareness and policy changes in the future in the discussion.

• There should be some background on how interactive teaching can be effective. There needs to be some citation on environmental education. For example, why did the authors choose interactive teaching session of 1 hour? Has education shown to be effective in changes behaviors?

Methods

• I am unclear as to why the authors chose 2 different subject to perform an interactive teaching session on. This is more testing if the study population cares more about vaccinations or air pollution not whether the interactive teaching method is effective. If the authors wanted to test that (as their title suggests) than the control group should either receive the information in a different way, such as only reading material, or none at all. Additionally, 8th graders may not have a real choice on whether they get vaccinated or not so this is a bit of an odd choice.

• The authors state that the teaching session was based on available literature but what available literature? Are they articles on air pollution, teaching methods, or both? There is also a lot of pre-existing lesson plans for environmental education. How do these sessions compare to those?

• Does the study population live in an area of high air pollution (I believe it does since they are in San Jose)? It would be useful to mention that as well as some summary statistics of air pollution levels and asthma rates or other childhood illnesses due to air pollution within the study area.

• Where did the questions for the questionnaire come from? It says from the research group but is there existing literature that shows that these questions are getting the information that the authors want. In other words, were this questionnaire previously validated?

•

• Is one month after the teaching session sufficient to show permanent behavior changes? Where did this time frame come from?

Results

• Again I am not sure that the study was designed in a way to test the effectiveness of this one-our teaching lesson.

• If this is mainly a hispanic population were the questionnaires also administrative in Spanish?

Discussion

• Is a one-hour teaching session really going to that effective? Would it not be better to design a longer course?

Reviewer #3: Note: There are no line numbers so comments provided below with ref to page numbers

SUMMARY

The study is interesting, and the manuscript is well-written, but more information on the development of the intervention – and especially on the development of the survey questionnaire and associated evaluation metrics (re impact on derivation of the MID used) - and associated limitations are needed for the reader to better evaluate the intervention and its evaluation. As presented, it is not clear that the statistically significant but apparently very small effect difference for the behavior assessment is in fact clinically significant (and reported pre and post differences in the control group could be interpreted to further indicate that it may not be clinically significant). The study has a number of limitations that are not sufficiently addressed and should be presented in the discussion section.

ABSTRACT

Background – revise statement re “daily living choices” – be more specific since this current sentence is not helpful, and also does not help improve understanding of claims in Abstract “Discussion” sentence re “env friendly behavior choices”

Objective – add for clarity “could shift REPORTED behavioral choices”

Methods – for added clarity add “ACTIVE control group”

- An important potential limitation is the metric used for evaluation since the survey was developed for this study but there is little info on this in the main text, for the abstract suggest making this clearer that the survey was new/designed for this study…

Results – perhaps another measure to report re income since range is so wide for median?

INTRODUCTION

P8 – clarify what is meant, or give a few examples parenthetically, re “personal choices in their daily lives” – same issue with second paragraph – and which populations are you targeting/referencing? USA specifically or high-income countries, pls clarify too

P8 – Sent beginning “One potential” – this is too broad of an assertion, suggest revising to “One potential FACTOR ASSOCIATED WITH the disconnect…”

METHOD

If this was a “pilot” study, make this clearer earlier on not just in the discussion (helps explain the relatively small n)

More info on HOW the one-hour teaching session was developed is needed – details can be provided in the SI (there is currently a link to slides, but this should be provided in the SI)

Given the centrality of the survey instrument to evaluation and outcomes it should be provided as part of the SI

P10 – How were the three classrooms selected? Provide more info/details – and to confirm: three classrooms of students from one school yes? Also re the timeframe was this study rolled out to each classroom simultaneously or otherwise? Explanation is needed generally and especially in light of the nearly four-year time period of the study (?)

P11 and 12 - Provide more info/details re “simple randomization” What was the breakdown re treatment/control by classroom following randomization?

P11 – So teachers facilitated recruitment and enrollment? If so state clearly, if not explain

P12 – explain “spirometry maneuver”

P12 – were any educators, or education researchers, involved or consulted in the creation of the teaching session – explain either way (if not, add to limitation sections)

P13 – was there any testing/evaluation of the survey-questionnaire (psychometrics)? Explain either way and if not add to limitations

P14 – assuming this evaluation instrument has not been used before, pls add some text to contextualize the use of such a tool and resulting scores/scales for this type of evaluation..

P14 – re regression, where any other variables other than age, sex, ethnicity available? – eg, income data is mentioned previously and reported in Table 1, why was this not also used? (later mentioned in the discussion but “data not shown”)

No mention of power calculations – pls add or add to limitations or both

Related, was there a pre-specified statistical analysis plan? If yes, what covariates were pre-specified? If not, add to limitations

RESULTS

Fig2 and P17/18 – given the potential limitations associated with the evaluation metric and range used, it is difficult to determine whether the small statistically sign difference is in fact clinically significant or not…

Based on the available information and data, I don’t think there is sufficient evidence to claim that these observed differences are a “significant and important increase” …more info, as requested above, about the evaluation instruments might help in this regard…

P18 – Again, why no adjustment for reported family income?

P18 – Suggest removing word “Remarkably” since it’s likely not appropriate given the design limitations and sample size…

Also the observed difference in the C group, if anything, adds contextual evidence that the observed difference in the T group discussed is likely not clinically significant… (re discussion: given the population and nature of the intervention, the assumption that this might have been due to spillover seems unlikely – but given the way the intervention was delivered it’s of course possible)

DISCUSSION

There are many methodological limitations that need to be more clearly discussed. The authors should create a dedicated Limitations section (as of now, limited discussion of limitations is somewhat scattered) or paragraph/s in the discussion section to address some the following and other points raised above:

Pls provide some info about the 25 students who declined ascent/consent and if/how they appeared to differ from those participating – if info not available pls state

Why conducted over almost four years if only three classrooms and short follow-up?

With such a relatively small n it may have made sense to use block randomization to better balance covariates (eg, income variable) – this is kind of mentioned re current discussion of cluster design (note that this would actually not be appropriate if only three classrooms) - should be discused with other limitations

Some discussion re difference between T and C groups for those <25k income should be added

One of the primary limitations – not unique to this study – is the use of a custom-designed survey and also study-specific scores/metrics to evaluate the outcomes of interest – both present substantial opportunities for bias – associated issues and limitations (re responses to questions above) should be discussed

P21 – As it stands, not enough evidence/info to support claim that a main strength of the study was “development of teaching materials and questionnaire” – indeed this could be construed as a primary weakness – more information is needed so the reader can better understand/decide

6. PLOS authors have the option to publish the peer review history of their article (what does this mean?). If published, this will include your full peer review and any attached files.

Reviewer #1: No

Reviewer #2: No

Reviewer #3: No

---

## [Author Response · Author response to Decision Letter 0]

25 Aug 2023

Please see the attached file under Response to Reviewers.

---

## [Decision Letter · Decision Letter 1]

28 Sep 2023

Effect of a single one-hour teaching session about environmental pollutants and climate change on the understanding and behavioral choices of adolescents: the BREATHE pilot randomized controlled trial

PONE-D-23-13463R1

Dear Dr. Arjomandi,

We’re pleased to inform you that your manuscript has been judged scientifically suitable for publication and will be formally accepted for publication once it meets all outstanding technical requirements.

Kind regards,

Timothy J Wade, Ph.D

Academic Editor

PLOS ONE

Additional Editor Comments (optional):

Please see two additional minor comments from Reviewer #1. You may consider addressing these briefly as you prepare the final manuscript.

Reviewers' comments:

Reviewer's Responses to Questions

**Comments to the Author**

1. If the authors have adequately addressed your comments raised in a previous round of review and you feel that this manuscript is now acceptable for publication, you may indicate that here to bypass the “Comments to the Author” section, enter your conflict of interest statement in the “Confidential to Editor” section, and submit your "Accept" recommendation.

Reviewer #1: All comments have been addressed

Reviewer #3: All comments have been addressed

2. Is the manuscript technically sound, and do the data support the conclusions?

Reviewer #1: Partly

Reviewer #3: Yes

3. Has the statistical analysis been performed appropriately and rigorously? 

Reviewer #1: No

Reviewer #3: Yes

4. Have the authors made all data underlying the findings in their manuscript fully available?

Reviewer #1: Yes

Reviewer #3: Yes

5. Is the manuscript presented in an intelligible fashion and written in standard English?

Reviewer #1: Yes

Reviewer #3: Yes

6. Review Comments to the Author

Reviewer #1: The authors have put in great effort to address the comments.

Minor comment(s)

For the skewed family income, the information on income level categories were derived and whether it is a standard category level or derived based on the data distribution etc is to be stated. Detail reason behind not to include the family income in the GEE is to be provided since GEE is quite a robust statistical test to some extent but needs to be cautious when comes to the interpretation of results.

The coding/labels for the variables e.g. nominal, non-binary is to be provided and whether dummy variables or machine learning approach were used in the regression analyses.

Reviewer #3: The authors have addressed all key comments in a satisfactory fashion and the manuscript is now much improved as a result.

7. PLOS authors have the option to publish the peer review history of their article (what does this mean?). If published, this will include your full peer review and any attached files.

Reviewer #1: No

Reviewer #3: No

---

## [Editor Report · Acceptance letter]

14 Nov 2023

PONE-D-23-13463R1 

Effect of a single one-hour teaching session about environmental pollutants and climate change on the understanding and behavioral choices of adolescents: the BREATHE pilot randomized controlled trial 

Dear Dr. Arjomandi:

I'm pleased to inform you that your manuscript has been deemed suitable for publication in PLOS ONE. Congratulations! Your manuscript is now with our production department. 

Kind regards, 

on behalf of

Dr. Timothy J Wade 

Academic Editor

PLOS ONE